



# Action-based flood forecasting for triggering humanitarian action

Erin Coughlan de Perez[1,2,3], Bart van den Hurk[2,4], Maarten K. van Aalst[1,3], Irene Amuron[5], Deus Bamanya[6], Tristan Hauser[7], Brenden Jongman[2,8], Ana Lopez[9], Simon Mason[3], Janot Mendler de Suarez[1,10], Florian Pappenberger[11], Alexandra Rueth[12], Elisabeth Stephens[13], Pablo Suarez[1,14], Jurjen Wagemaker[15], Ervin Zsoter[11]

[1] Red Cross Red Crescent Climate Centre, The Hague, 2521 CV, The Netherlands
[2] Institute for Environmental Studies, VU University Amsterdam, 1081 HV, The Netherlands
[3] International Research Institute for Climate and Society, Columbia University, New York, 10964, USA
[4] Royal Netherlands Meteorological Institute (KNMI), De Bilt, 3731 GA, Netherlands
[5] Uganda Red Cross Society, Kampala, Uganda
[6] Uganda National Meteorological Authority, Kampala, Uganda
[7] Climate System Analysis Group, Department of Environmental and Geographical Science, University of Cape Town, Cape Town, South Africa
[8] Global Facility for Disaster Reduction and Recovery (GFDRR), World Bank, Washington DC
[9] Atmospheric Oceanic & Planetary Physics Department, Oxford University, OX1 3PU, United Kingdom
[10] Frederick S. Pardee Center for the Study of the Longer-Range Future, Boston University, Boston, Massachusetts
[11] European Centre for Medium-Range Weather Forecasts, Reading, RG2 9AX, United Kingdom
[12] German Red Cross, Berlin, 12205, Germany
[13] School of Archaeology, Geography and Environmental Science, University of Reading, Reading, RG6 6AH, United Kingdom
[14] Department of Science, Technology, Engineering and Public Policy, University College London
[15] Floodtags, The Hague 2516 BE, The Netherlands

*Correspondence to*: E. Coughlan de Perez (coughlan@climatecentre.org)

**Abstract.** Too often, credible scientific early warning information of increased disaster risk does not result in humanitarian action. With financial resources tilted heavily towards response after a disaster, disaster managers have limited incentive and ability to process complex scientific data, including uncertainties. These incentives are beginning to change, with the advent of several new Forecast-based Financing systems that provide funding based on a forecast of an extreme event. Given the changing landscape, here we demonstrate a method to select and use appropriate forecasts for specific humanitarian disaster prevention actions, even in a data-scarce location. This action-based forecasting methodology takes into account the parameters of each action, such as action lifetime, when verifying a forecast. Forecasts are linked with action based on an understanding of (1) the magnitude of previous flooding events and (2) the willingness to act "in vain" for specific actions. This is applied in the context of the Uganda Red Cross Society Forecast-based Financing pilot project, with forecasts from the Global Flood Awareness System (GloFAS). Using this method, we define the "danger level" of flooding, and we select the probabilistic forecast triggers that are appropriate for specific actions. Results from this methodology can be applied globally across hazards, and fed into a financing system that ensures that automatic, pre-funded early action will be triggered by forecasts.



## 1 Introduction

Taking preparedness actions in advance of a disaster can be both effective in saving lives and assets, as well as efficient in reducing emergency response costs. Practitioners and forecasters have mobilized around the concept of "Early Warning Early Action" based on weather information (Alfieri et al., 2012; IFRC, 2009; Krzysztofowicz, 2001; Webster, 2013), also in light
of rising risks in a changing climate (e.g., IPCC, 2012). In this context, there is considerable demand for decision-relevant climate and weather information. The humanitarian and development sectors collaborate with forecasters on early warning for disaster risk reduction, for instance in the context of the Global Framework for Climate Services (Hewitt et al., 2012) and the regional Famine Early Warning System Network (Ross et al., 2009).

Disaster managers have indeed been highly successful in using forecasts in cyclone-prone areas of the world: actions based on early warning systems have saved millions of lives and prevented significant damage (Galindo and Batta, 2012; Harriman, 2013; Lodree, 2011; Rogers and Tsirkunov, 2013). This is partly because people can take action when they know that a cyclone is nearly certain to strike, and cyclones can have enormous impact on society. In addition to cyclones, heat wave early warning systems also trigger action to reduce mortality; these are most commonly established in developed countries (Ebi et al., 2004;
Fouillet et al., 2008; Knowlton et al., 2014a).

These advances contrast sharply with the systematic lack of humanitarian action before other predictable natural hazards, including flooding. The barriers to early action are particularly apparent in data-scarce areas of the developing world (Brown et al., 2007; Houghton-Carr and Fry, 2006).

One major barrier is the lack of funding available when a disaster is likely, but not certain. This incentive structure is beginning to change with the advent of new Forecast-based Financing systems (Coughlan de Perez et al., 2015). These systems allocate resources *prior* to a hazard occurring based on a preselected forecast. This allows for the possibility of acting "in vain" if the hazard does not occur, ensuring that the long-term gains of preventative action will outweigh the costs of false alarms. Here,
we explore two specific challenges for the development of such a system in the context of a probabilistic flood forecast, and offer a forecast evaluation methodology tailored to specific actions. This builds on existing methodologies to match forecasts with actions in light of the costs and benefits of these actions (Coughlan de Perez et al., 2015; Lopez et al., n.d.).

First, translating flood magnitudes into damages is a non-trivial task in a data-scarce location. Dale et al., (2012) proposed a
method to convert forecast probabilities from an ensemble system into likelihoods of damages using a magnitude-damage curve, aggregated proportionally by each ensemble member. However, the data requirements of creating such stage-damage curves (Merz et al., 2010; Michel-Kerjan et al., 2013; Ward et al., 2013) are often prohibitive, as the precise amount of flooding



that will cause impact is often unknown. Here, we offer an alternative methodology to identify the critical flood magnitude that needs to be forecast to inform humanitarian action.

Secondly, flood forecasts, especially in data-scarce areas, have high uncertainties. While there may be demonstrable
probabilistic skill in flood forecasts (Alfieri et al. 2013), probabilities themselves open the possibility of action "in vain". Here, we consider action "in vain" as action that is taken after a forecast, but is not followed by the extreme event. In such a case of action "in vain", the humanitarian actor would have chosen an alternative use of resources if he/she had known that the extreme event would not materialize.

Therefore, humanitarian actors are often unsure of when it would be *worthwhile* to take action and spend resources based on a probabilistic forecast. Analyses of prepositioning of stocks rarely consider how forecast probabilities could be used to trigger such action – or "action-based forecasting" (Bozkurt and Duran, 2012; Bozorgi-Amiri et al., 2011). Without a confident answer that links specific actions to specific forecast probabilities, disaster managers find themselves immobilized in discussions at the moment of receiving a forecast of likely extreme conditions, with little criteria or clarity on how to make a decision and
take action.

Hence, the aim of this paper is to develop a methodology to link together forecasts and appropriate humanitarian actions; in doing so, we acknowledge the challenge of using forecasts in data-scarce areas. Specifically, we address two questions:

1. Given limited observational data and historical forecasts, how should the hydrometeorological *danger level* threshold
that represents an impactful flood be chosen?
2. Given the limitations of assessing forecast skill using limited observational data, how should the forecast probability to *trigger* early action be identified?

In this paper, we illustrate the practical application of this methodology for a pilot Forecast-based Financing project in rural
Uganda. We evelute river discharge forecasts from the Global Flood Awareness System (GloFAS), a global hydrological model run daily using rainfall forecasts from the European Centre for Medium Range Weather Forecasting (ECMWF). After introducing the context of the project region, we elaborate a method for selecting the danger level and trigger, including constraints that need to be included to ensure the method is applicable to a humanitarian situation. We then share results from two locations in North Eastern Uganda, and estimate the probability that a system predicated on such limited data will be
"intolerable", or cause disaster managers to act "in vain" more often than was expected. Based on this, we discuss implications for North Eastern Uganda and other regions. We conclude with proposed next steps for Forecast-based Financing systems, and application of global flood models elsewhere.



## 2 Context

### 2.1 Region

The Uganda Red Cross Society, with support from the German Red Cross and the Red Cross Red Crescent Climate Centre, is implementing a Forecast-based Financing pilot in the North Eastern part of the country. As part of this pilot, the German Red

Cross established a novel Preparedness Fund that can be disbursed to take predefined preparedness actions when a triggering forecast is issued in this region. At the time of writing, there are more than a dozen such Forecast-based Financing projects operational globally.

The Teso region of North Eastern Uganda is a swampy region, prone to river flooding and waterlogging during the two rainy

seasons centred in May and October. The Uganda Red Cross Society project areas are in the sub-districts of Magoro and Ngariam in Katakwi district on the Apapi River, and Kapelebyong in Amuria district on the Akokoro River. Unfortunately, there is no calibrated hydrological model available for these rivers. Both rivers drain into Lake Bisina and eventually into the Nile.

*[Figure 1 see end]*

The Uganda Red Cross Society selected this pilot region based on vulnerability to floods. As regional conflict subsided in the 1990s and 2000s, this region was gradually re-settled, and nowadays many of the current residents practice farming and raise livestock. Since that time, several flood events have impacted the area. The floods typically cause impassable roads, loss of

crops, outbreaks of waterborne diseases, and collapse of houses and latrines (OSSO and LA RED, 2009).

Whenever a flood is reported, Uganda Red Cross Society has a mandate to assess the situation and respond. In past events such as the 2007 floods, they have provided post-disaster shelter and relief items to the affected population (Jongman et al., 2015). Both the flood losses and the disaster response expenses could be reduced if anticipatory measures were deployed

before the flood, after unusual conditions are forecast. Based on the methodology articulated in this paper, Forecast-based Financing thresholds were operationalized in mid-2015, consisting of Standard Operating Procedures for Forecast-based Action. In November 2015, a "triggering" forecast successfully initiated action (Red Cross Red Crescent Climate Centre, 2015). This was the first time the local branch had used a Preparedness Fund to take action before flood disaster reports were issued.

### 2.2 Actions

To set up the Forecast-based Financing system to initiate early action, the Uganda Red Cross Society project team identified preparedness actions that could be taken prior to a flood event, through consultations with people living in the affected areas





as well as internal discussions and two facilitated workshops. Participants at the workshops included disaster managers, volunteers, the Uganda Meteorological Authority and district officials (Jongman et al., 2015). In each of the workshops, the disaster managers from Uganda Red Cross Society discussed the quantitative and qualitative costs and losses associated with three scenarios: (1) taking successful action, (2) failing to act before a flood, and (3) acting "in vain". For each action, they first answered questions individually before discussing collectively. Lastly, disaster managers estimated their willingness to "act in vain", expressed as a number of times out of 10.

Ultimately, the team selected a set of actions that were seen as both impactful and implementable by the Uganda Red Cross Society. One action (evacuation), was eliminated because about one-quarter of the respondents indicated that they would not be willing to act in vain at all. The political and reputational costs of evacuating in vain are considerable. The remaining selected actions are specified in Table 1. For these three actions, the disaster managers came to a consensus that they would be willing to act in vain approximately 50% of the time. Here, we use this as the "tolerable" amount of acting in vain to establish the Forecast-based Financing system for this set of actions. Later in the paper, we estimate the probability that the GloFAS forecast triggers are an "intolerable" system, or one that causes disaster managers to act "in vain" more than 50% of the time.

For each action, the Uganda Red Cross Society specified how many days would be needed to carry out the action, which should correspond to the forecast lead-time (Jongman 2015). The specified lead-times are contingent on the assumption that several of the procurement and volunteer training steps would be carried out at the beginning of the flood season, to enable quick action based on a short-term forecast.

Secondly, they identified the "action lifetime"; the period of time after the action is completed during which it offers preparedness or protection from the extreme event. Traditional flood forecast evaluations are specific to the time period forecasted, evaluating whether a single forecasted day did indeed flood. Humanitarians would count this as a "hit" and, unlike forecasters, they would also consider it to be a "hit" if the flood instead occurred 5 days after the forecasted date and the action lifetime was 30 days: In such a case, the action would still be effective in reducing impacts, even though the flood occurred slightly later than the forecasted date.

Therefore, the methodology detailed in this paper avoids re-triggering an action if the "action lifetime" of a previous action is still ongoing. For example, after digging drainage, the team would not re-trigger digging of trenches until the first set of trenches could be assumed to have degraded, likely about 90 days after digging. While the end of the "lifetime" is not a strict transition from useful to non-useful, it is an estimation of the date at which the Uganda Red Cross Society would find it acceptable to re-trigger the action in the region. We posit this constraint throughout the paper; an action cannot be re-triggered until the action lifetime of the preceding action is over.



The selected actions are listed in table 1 (note that this is subset of all actions that were originally considered).

| Action | Time required to complete the action (implementation time) | How long the action will benefit the community after it is completed (action lifetime) |
|---|---|---|
| Water storage and purification: distribute jerry cans, soap, and a 30-day supply of chlorine tablets to vulnerable households | 4 days to complete | 30 days after completion |
| Water drainage: dig trenches around homes to divert water | 4 days to complete | 90 days after completion |
| Food Storage: bag vulnerable items and move to storage facilities on high ground | 7 days to complete | 30 days after completion |

*Table 1: Actions selected for a Forecast-based Financing system. See* (Jongman et al., 2015) *for more information on the actions and their associated costs. Note that the implementation time of an action should equal the lead-time of the forecast selected to trigger that action.*

### 2.3 Forecasts

This paper proposes a method for identifying a forecast that could trigger one or more of these actions before a potential flood, given the constraint of acting in vain less than 50% of the time. As mentioned earlier, there is no locally available flood forecasting system, and there is only one river gauge with recorded discharge in the pilot area. Unfortunately, the large upstream catchment area dictates that rainfall in a specific village is not a useful proxy for flood risk in that village. Given these constraints, we choose to examine whether river discharge forecasts from the Global Flood Awareness System (GloFAS)

can be used to trigger action in this data-scarce location in ways that are compatible with stakeholder priorities. Probabilistic hydrometeorological forecasts have been evaluated globally, and have been shown to have limited skill (e.g. Alfieri et al., 2012; Li et al., 2008; Wu et al., 2014).

GloFAS is an operational global ensemble flood forecasting system developed in partnership between ECMWF, the European

Commission Joint Research Centre, and the University of Reading (Alfieri et al., 2013). Currently in a pre-operational development phase, calibration of the model with river flow observations, where available, is being carried out in a research mode. The model version used here is not calibrated for the North Eastern Uganda catchments. GloFAS is run once a day to produce probabilistic discharge estimates over the entire globe at a resolution of 0.1 degrees (approximately 11km at the equator). Here, we use daily historical GloFAS forecasts from 2009-2014, as well as gauge data from the (the only) local

Akokoro Gauge from 2009-2013, which overlap for 2,014 days. The gauge is located at approximately 1.86N, 33.85E.





GloFAS is driven by the ECMWF ensemble forecasting system, with 51 ensemble members at lead-times of 0 to 45 days. The first 15 days include rainfall forecasts, and the following days are river routing only. The probabilistic flood forecasts are available free of charge on a password-protected website (http://www.globalfloods.eu/). GloFAS takes a 'model climatology' approach, aiming to forecast extremes or anomalies in river flow relative to historical 'climatology' runs of the model (Hirpa et al., 2016). This approach addresses the problems of the lack of representation of local scale channel geometry and bias in the precipitation forcing. However, one of the major challenges is to link the model climatology to the real world, focusing on the percentiles rather than absolute values of the forecast.

## 3 Methods

To define a forecast probability that could be used to trigger early action in the Uganda Forecast-based Financing system, we (1) estimate the quantity of discharge that represents a "flood" and (2) identify the forecast probability that will make it worthwhile to take preparedness actions (less than 50% chance of acting in vain).

*What hydrometeorological "danger level" represents a flood?*

While the relationship between water levels and flooding will vary over time due to trends in vulnerability and exposure, here we define a percentile of discharge that is qualitatively associated with reported flood events of the past few years, when avoidable losses were observed. For flood reports, we use two sources of information: humanitarian records and media reports.

With regards to the humanitarian records, we combine records from the Desinventar database (UNISDR 2011) with an internal record system of disasters that are reported to the Uganda Red Cross Society. Between the two humanitarian datasets, they contain 8 distinct records of floods in the Magoro area from 2009 to mid-2014; these floods occurred in 2010, 2011, and 2012.

For the media analysis, we analysed two national Ugandan newspaper repositories: Daily Monitor and New Vision. We filtered each repository on 40 flood-related keywords[1]. From Daily Monitor we downloaded a total 2974 news articles between 2004 and 2015. From New Vision we downloaded 752 news articles between 2001 and 2015. Unfortunately the database for New Vision could not be fully accessed, since the news repository allows access to only the top 200 newspapers per query, without possibility for advanced search.

---

[1] The keywords are: flood, floods, flooding, inundation, inundations, landslide, dam break, dam burst, dam bursting, dam breached, dam fail, dam failed, dam failing, dam failure, dam broken, dam collapse, dyke break, dyke burst, dyke bursting, dyke breached, dyke fail, dyke failed, dyke failing, dyke failure, dyke broken, dyke collapse, embankment break, embankment burst, embankment bursting, embankment breached, embankment fail, embankment failed, embankment failing, embankment failure, embankment broken, embankment collapse, submerged, overflowed, breach, water-logging.



Within the database total of 3726 news articles, we clustered the sentences in the articles using a K-means clustering algorithm. Next we annotated the clusters using four classes: 1. Current flood event 2. Past event or flood warning 3. Mixed and 4. Unrelated. After annotation we found that a total of 1721 news articles held relevant flood information (annotated as class 1 or 2). To obtain geographical information, we filtered the sentences for terms that are often related to a location[2] and within this subset looked for mentions of district and sub-county names. As a result, for the district of interest (Katakwi) we found a total of 27 news articles with flood sentences AND geographical reference. Applying the same approach to all districts in Uganda we found a total of 1173 of such articles (except in this case we did not only use the sentences containing geographical related keywords).

With these results from the algorithm, we validated the result manually for the districts of our interest by reading the articles. For 85% of the events we had found an actual flood event, meaning that the flood event was automatically detected for the correct month/ year in the correct location(s). Conversely, 15% were false positives. The result of this data mining of the news repositories is a historical flood overview with dates of flood occurrences in Katakwi district (it can be accessed here: https://www.floodtags.com/historic-floodmap-uganda/). There are 13 newspaper reports of flooding within our timeseries.

While this accounts for many events, not all disasters are included in these databases, and some of those included may have had less impact than others. The effect of this under-representation is an overestimation of acting in vain, which renders our trigger selection conservative. In addition, impact is not perfectly correlated with flood magnitude, given that vulnerability can change over time. Therefore, we only attempt a qualitative comparison of discharge and reported flood events, which adds additional (unquantified) uncertainty to the calculation of false alarms in the following section.

As we do not have a gauge for the Apapai River, where Magoro and Ngariam sub counties are located, we use the daily ensemble median of GloFAS forecasts at lead time of 0 as a proxy for actual discharge and compare this with the above datasets of reported disasters in those two locations. We qualitatively select a threshold percentile of discharge to be considered the "danger level" or "flood" for this region, rather than an absolute value. The exact percentile is a subjective selection to approximate the base rate of reported floods, ideally including the maximum number of exceedances that were indeed followed by a reported flood event.

*What forecast probability should trigger action?*

---

[2] They are: affected NOT not, hit NOT not, situation AND bad, situation AND worse, situation AND worst, cut off, displaced, destroyed, submerged, collapsed





Using this selected "danger level" percentile as a proxy for the amount of discharge that causes a flood, we calculate what probability of exceeding the danger level should trigger action. Here, we calculate probabilities using the forecast ensemble, evaluating them against the gauge discharge.

The forecast verification score of interest to humanitarian actors is the False Alarm Ratio (FAR) (Hogan and Mason, 2012; Lopez et al., n.d.), defined as the number of forecast-based actions that were *not* followed by a flood, divided by the total number of actions that were triggered by the system. It thus represents the proportion of actions that are taken "in vain". Here, we take into account the action lifetime, so any action that was followed by a flood during its lifetime is considered a "hit", and only actions that have no flooding at any point during their lifetime are "in vain". Similarly, a second action is never
triggered during the lifetime of another action.

To estimate the FAR, we compare the nearest gridbox of the GloFAS forecast with the river gauge on the Akokoro river, for which we have an overlapping time period of daily data from 10 January 2009 to 31 December 2014 The correlation of these two datasets is 0.52. In the context of a Forecast-based Financing system, the Red Cross or other humanitarian actors will take
action when the forecast *meets or exceeds* a triggering probability of flooding. The FAR is therefore calculated as follows: (1) any forecast meeting or exceeding the trigger probability is considered an action (2) any action followed by a flood within the action lifetime of 30 days is counted as a "hit", otherwise an action "in vain".

Our first goal is to estimate whether a forecast indicating a 50% chance of flooding would indeed correspond to a 50% chance
of flooding in the real world. We plot reliability diagrams (Broecker, 2012) for the forecast at 4-day and 7-day lead times at the gauge location, as well as the GloFAS forecasts in the two non-gauge project locations, comparing 4-day lead time forecasts with the 0-day forecasts to approximate actual discharge. However, in such a small sample, the incidence of forecasts of rare events is low, and therefore the confidence intervals on these reliability diagrams are very wide.

Given such a small number of years to calculate the performance of forecasts with regards to extreme events, however, we cannot be sure that the estimate of FAR in the sample is representative of the true value. For example, if the estimate from our sample yields a FAR of 30%, it is still possible that the real value is actually greater than 50%. This means that, in reality, the selected trigger level for our Forecast-based Financing system would cause the Uganda Red Cross Society to act "in vain" more than 50% of the time, which is not considered "tolerable". To estimate the risk of setting up an "intolerable" system, we
calculate confidence intervals around the FAR by using bootstrap resampling. To account for the autocorrelation of the discharge time series, we use a 60-day fixed block bootstrap to generate 10,000 samples by resampling with replacement the time series (n=2014) of forecast-observation pairs. Given a trigger forecast probability, for each sample we calculate the FAR and generate a distribution of all the sample FARs. This is repeated for each trigger probability, and we demonstrate results for three triggers: forecast probabilities of 30%, 50%, and 70%. Based on these results, we estimate the likelihood that taking





action when one of these forecast probabilities is exceeded will lead to a FAR above 50%, which would fail to satisfy the decision-maker requirements for action "in vain".

## 4 Results

*What hydrometeorological danger level represents a flood?*

To estimate the percentile of discharge that is associated with flooding in the project region, we plot the historical median water levels forecasted at lead time of 0 by the GloFAS model. Here, we focus on the Apapi River, where two project districts are located and several disaster records exist. Because Ngariam sub county is directly upstream of Magoro sub county, we plot simulated discharge at Magoro and reported flood events in both sub counties. Comparing this with historical floods (dark blue

lines), and media reports from the district (light blue lines) we qualitatively select the 95th percentile (horizontal red line) as a proxy for disaster.

*[Figure 2 see end]*

In the 6 years of 2009-2014, this danger level would have been exceeded in 2010, 2011, and 2012. In April 2010, reports indicate that 12 secondary schools and 7000 people were affected by flooding in the area, followed by crop losses due to waterlogging in May 2010. Flooding continued to be reported through September and October of that year, affecting several regional roads. This corresponds well with the simulated discharge for those years.

In 2011, simulated discharge again accords well with reported flooding that affected both people and infrastructure in the area. In 2012, waterlogging reports to Uganda Red Cross Society arrived in August, which is substantially after the peak modeled discharge, and the newspaper reports are concentrated in October and November. It is possible that the peak discharge did correspond with the model data and was not reported, or was reported at a later time. Our threshold was not crossed in 2013 and 2014, which accords well with the lack of reported floods in those years.


We begin to see other years (with no disasters) counted as "floods" if we lower the danger level below 93%, and if we raise it above the 99th percentile very few years exceed the threshold. Therefore, we assume from the limited data available that discharge above the 95th percentile is likely indicative of flood conditions in this location, and in the following analysis, anything above the 95th percentile is defined as a "flood". In Kapelebyong sub county, the other project location in this region,

the only recorded disasters are from the devastating floods of 2007, which are not available in GloFAS reforecasts. Therefore, we also assume this percentile applies to Kapelebyong, as the infrastructure and vulnerability are similar in the two areas.



*What forecast probability should trigger action?*

We consider forecasts of 4-day and 7-day lead-times, aiming to identify a trigger that corresponds to a FAR of 0.5 or less. If the forecast probability of exceeding the flood danger level is 50%, then the observed frequency of exceeding the flood threshold should be 50% for a reliable forecast.

*[Figure 3 see end]*

In Figure 3, we plot the reliability of the forecast at both lead-times when compared to gauge discharge on the Akokoro River. In the project locations with no gauge, we also examined the ability of GloFAS to forecast itself 4 days in advance (Figure 3,

right-hand reliability diagram). In both cases, we are unable to establish the reliability with confidence given the small sample size.

If we set the trigger given this limited data, how likely is it that we developed a system that is "intolerable" to the Uganda Red Cross Society, actually leading disaster managers to act "in vain" more than 50% of the time? Figure 4 shows the FAR from

10,000 resamples as a probability distribution function. This assumes that action is triggered when the forecast probability of flooding reaches or exceeds a 30%, 50%, or 70%, and that there is a 30-day action lifetime.

The bootstrapped results indicate a high chance of a "tolerable" system, especially at higher forecast triggers. Only 24%, 19% and 18% of all the bootstrapped samples returned an "intolerable" system (grey bars) for a threshold of 30%, 50%, and 70%,

respectively. This is true for a sample size of 2,014 days. This represents the chance that the system does not pass the required specifications, and would cause humanitarian actors to act "in vain" more than 50% of the time in the long run. While increasing the forecast trigger does reduce this risk, the effect is not substantial given the small dataset available.

*[Figure 4 see end]*

**5 Discussion**

The calculations and estimations used here build on established forecast verification methods, combining information on both actions and forecast skill to enable the use of forecasts by the humanitarian community. Without incorporating information about the action, it is unlikely that the humanitarian community would be willing or able to plan for preparedness actions using existing seemingly-arbitrary forecast verification measures.




As illustrated here, there are two major components of action-based forecasting. Forecasters and disaster managers (1) select the appropriate danger level of a hazard that causes avoidable losses and (2) calculate the FAR for specific trigger probabilities based on willingness to act in vain.

These two components can be readily applied to most other forecastable natural hazards. First, there are many possibilities for defining the "danger level" of river floods both spatially and temporally; Stephens et al. (2015) has suggested several different definitions of "floodiness" that could correspond to danger levels in varying river situations. Outside of riverine flood hazards, "danger levels" of rainfall are available for flash flood events (Bacchini and Zannoni, 2003; Yang et al., 2015). Beyond floods altogether, heat waves are an example of hazard where there are many epidemiological studies to identify temperatures that
are linked to increase morbidity and mortality (Hajat et al., 2010; Knowlton et al., 2014b; WMO and WHO, 2015). The same applies to storm surge heights, drought indices, wind speeds, etc (Muir Wood et al., 2005; Ross et al., 2009).

Although the methods for defining the "danger level" for each type of hazard can and do differ, many do rely on reports of historical disaster events (Bacchini and Zannoni, 2003; Loughnan et al., 2010). The method of news repository data mining
used here is a scalable method to identify approximate dates of impacts. This can be enhanced by improving the geocoding database, (e.g. correcting errors in the OpenStreetMap database for Uganda), improving the clustering methods (e.g. isolating different flood incidences including blocked roads and improving geocoding) and negotiating access to more newspapers (e.g. better access to the New Vision repository). Further qualitative research on the news articles related to flooding in the region of interest can also help guide the selection of what types of forecast-based action would be most appropriate for the region.

The second component of action-based forecasting is calculation of the FAR at the specified danger level. Instead of static forecast verification metrics, the FAR for any hazard forecast should be calculated according to these context-specific parameters, including the action lifetime. The World Meteorological Organization has issued guidance on impact-based forecasting, which includes information on selecting threshold danger levels that then can be forecasted for target recipients
(WMO, 2015). This guidance does not address probabilities, using deterministic terminology such as "winds are expected". It does not include information on how to select trigger probabilities for a specific action, and could therefore be complemented by the techniques described here.

Critical to selection of triggers based on the FAR results is the estimation of willingness to act "in vain". When it comes to the
risk of an "intolerable" system that has too many false alarms, donors will also need to consider the implications of such a risk for their portfolio. In the Uganda example, disaster managers estimated that they would be willing to act "in vain" 50% of the time. It should be noted that there is evidence that when people are asked to express probabilities, their choice of 50% is often an expression of not being sure as to the answer (Fischhoff and Bruine de Bruin, 1999; Tetlock and Gardner, 2015). While the 50% constraint from local stakeholders was respected in this study, further research into decision science could improve how



this answer is elicited. Such research, in collaboration between forecasters, users, and behavioural scientists, could identify any biases that humanitarian decision makers should actively avoid.

Almost all of the steps in this analysis contained unquantifiable uncertainties. On the side of the forecasts, uncertainties can be
reduced with longer reforecast timeseries for each model update, as well as the implementation of local record-taking devices for calibration. On the side of the actions, uncertainties are likely larger and much more difficult to quantify. The lives and vulnerabilities of the people living in the target villages are constantly evolving, as are the capabilities and priorities of the humanitarian sector. While these are difficult to reduce, continual updates to danger levels and triggers as well as simulations with all relevant actors can confirm that the critical values and assumptions still hold.

**6 Conclusion**

Forecast-based Financing aims to link forecasts to actions in advance of disasters. In this applied research, we have illustrated the development of such a system in a vulnerable context, where calibrated local forecasts do not exist to support such decision-making. Examining the application of Forecast-based Financing in a data-scarce region of Uganda, we have proposed an action-based forecasting methodology to answer two critical questions to enable early action based on flood warnings:

1.   Given limited observational data and forecast availability, how should the *danger level* threshold be chosen that represents an impactful flood?

    2.   Given the limitations of assessing forecast skill using limited observational data, how should the forecast probability to *trigger* early action be optimized to avoid intolerable levels of false alarms?

Using this action-based forecasting methodology, we demonstrate that global flood products can already trigger worthwhile actions, even in data-scarce locations. Assuming that the 95th percentile of forecasted discharge is a valid proxy for a "danger level" in an area with limited data records, the GloFAS model can be used to trigger timely humanitarian action in advance of an extreme event. Not only is there early action that can be justified based on the False Alarm Ratio of the GloFAS forecast in this area, the probability of triggering an unacceptable level of false alarms is less than 25% in this region. Part of the reason
for such skill in the model is that the actions taken by humanitarians have long "lifetimes", and therefore are forgiving if the forecast is early and the flood comes late.

It is encouraging that a global flood forecasting system has the potential to support decisions, although this is not a replacement for better observational data or the development of calibrated catchment-scale models. While this method can successfully
forecast many instances of extreme river flow, it is only able to trigger actions that can in practice withstand a large number of false alarms. Indeed, better observational datasets and catchment-scale models could enable us to estimate the hazard-damage curve in a specific locality, modeling the precise level of discharge that causes inundation and associated impact in





specific areas. This type of modeling could allow for the selection of more specific and targeted preparedness actions, including actions specific for "small" floods that would no longer be useful in a major flood (e.g. storage of water, which would not be useful if one needed to later evacuate). Similarly, forecast-based actions could be crafted for different "types" of flooding, such as long-duration or single high flows (Stephens et al., 2015). Data constraints are often cited as the barrier to forecast-based

action in rural areas of the world, and longer data sets over time will indeed allow for more precise calculation of flood thresholds and the inclusion of additional triggers for action.

Model changes are continually implemented in real-time forecasting systems, and the experimental GloFAS model version used for this study has already been updated several times. These dynamic changes to the forecasting system add additional

uncertainty to the implementation. In each model update, the danger level and triggers need to be recalibrated with additional reforecasts, to assess how the danger level and FAR might have increased or decreased in a given Forecast-based Financing project location. To partly avoid this cumbersome requirement of constant reforecasts, forecasters developing an operational product can consider forecast corrections to ensure that the climatology of the model does not change with model updates. This will ensure that the danger level stays constant even if the FAR does change.

However, "the perfect should not be the enemy of the good". With relatively limited local and global data available, effective humanitarian action can be triggered in advance of potential flooding. Humanitarian actors have a mandate to serve vulnerable people, and cannot wait to engage in flood preparedness measures until sufficient local data is collected over the years to establish 'conventional' predictive models, especially when global models may give signals of likely extreme conditions in

the foreseeable future. Moreover, the Forecast-based Financing system based on this method of analysis did indeed trigger action for the first time in Uganda in November 2015, when water purification tablets, soap, shovels, and storage bags were distributed to the at-risk population. Evaluation of the entire system, including the effectiveness and timeliness of these actions, is ongoing.

This simple methodology can easily allow for improvement over time, adjusting parameters such as danger levels or probability thresholds as experience reveals to stakeholders' the desirability of redefining parameters based on objective calculations or valid subjective preferences. Additionally, this approach can be extended to other locations, and potentially scaled up to regional or national mechanisms that systematically trigger early action to address flood risks among vulnerable people around the world. In the long-term, there are opportunities to reduce the lead-time needed for preparedness measures, to offer more

choices of actions that can be taken in the window of time between a forecast and the potential disaster. Such innovations include everything from unmanned aerial vehicles to rapidly deliver health materials to rural locations (Bamburry, 2015) to blockchain and smart contract technologies allowing instant transfers of programmable money (Currion, 2015; Forte et al., 2015).



Operationalizing Forecast-based Financing systems is within reach. First, more flexible humanitarian financing is needed that allows and incentiveses early action despite the risk of acting in vain; this is currently being considered by various humanitarian and development donors. Successful implementation of such funding requires improved incentives towards early action and enabling an iterative learning process toward more effective links between early warning and early action. To achieve this,
further investments are needed at the practitioner interface between scientific and humanitarian organizations. Humanitarian actors need to identify risk reduction and preparedness actions that can be taken before a potential flood, and agree on their level of willingness to act "in vain" for each action. By the same token, natural scientists (e.g. forecasters and flood modelers) need to continue deepening their engagement with humanitarians and other stakeholders who can help turn scientific knowledge and skills into societal benefits. There is also a need for decision science expertise, to advance the design of
processes that can help to better engage stakeholders in understanding and defining thresholds. Further collaboration among researchers and practitioners on the development of such systems can unlock the potential to greatly reduce the consequences of recurrent disasters around the world.

**Acknowledgements**

The authors of this paper would like to extend their gratitude to the donors who have made the Forecast-based Financing pilots possible. The pilot project in North Eastern Uganda is funded by the German Federal Ministry for Economic Cooperation and Development (BMZ). In addition, the German Federal Foreign Office has developed an Action Plan for humanitarian adaptation to climate change (Rüth, 2015), and has invested in Forecast-based Financing in several other countries in Africa and around the world.
We would also like to thank many who have contributed to this approach, including Leo Mwebembezi of the Uganda Hydrological Department, the entire Uganda Red Cross Society implementing team, the GloFAS team, and participants in the Forecast-based Financing Dialogue Platform hosted biannually by IFRC in Geneva.

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



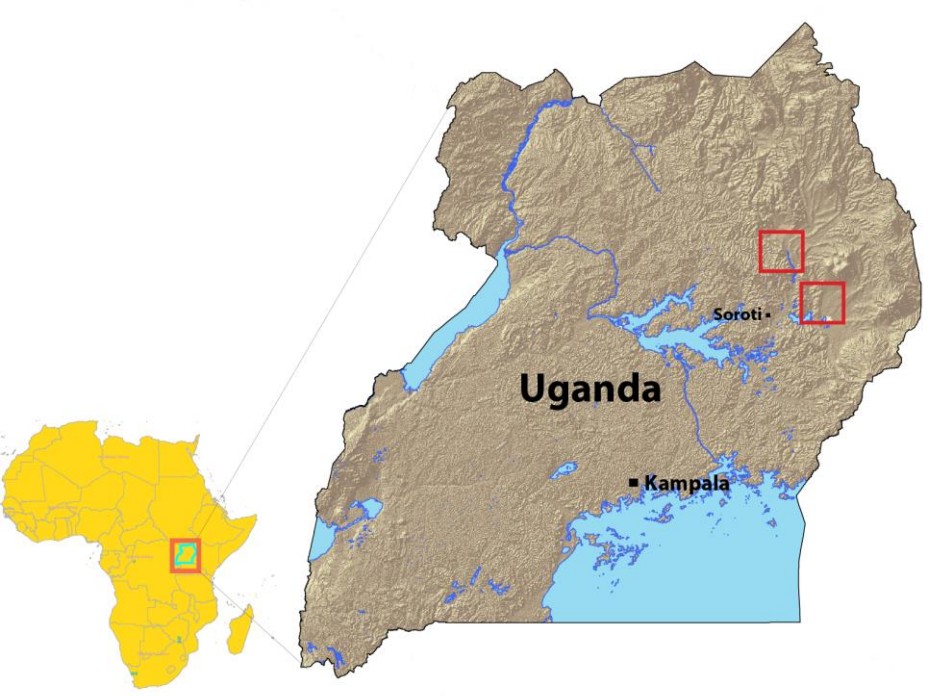

*Figure 1: Map of Uganda; Kapelebyong and the gauge are marked by the top red square, and Magoro and Ngariam are located at the bottom red square.*



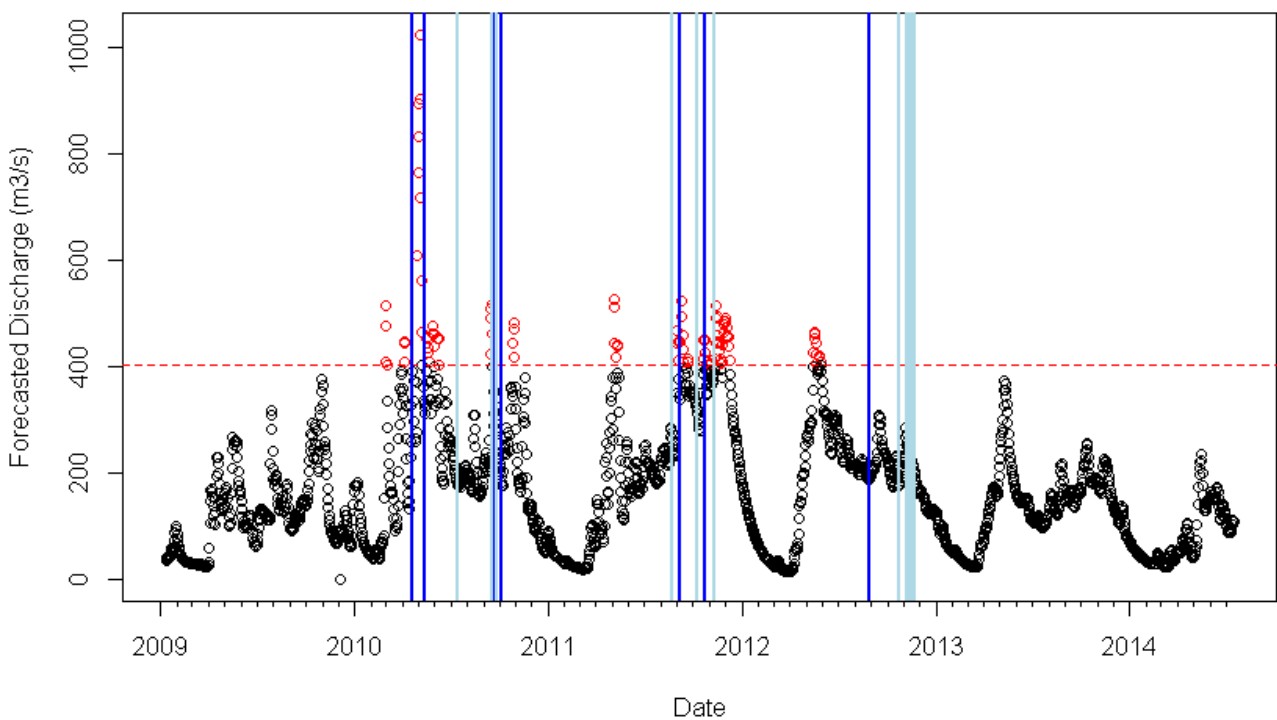

*Figure 2: Forecasted discharge (circles) at Magoro sub county, Uganda represented by GloFAS forecasts ensemble median at lead-time 0. Dates of disasters in the regions along the Apapai river are indicated by dark blue vertical lines, as per the databases of Uganda Red Cross Society and Desinventar. Newspaper reports of flooding in the district of Katakwi are indicated by light blue vertical lines. Small tick-marks on the x-axis correspond to months within a year. The horizontal dashed red line indicates the 95th percentile of estimated discharge; dates with discharge above this threshold are colored in red.*





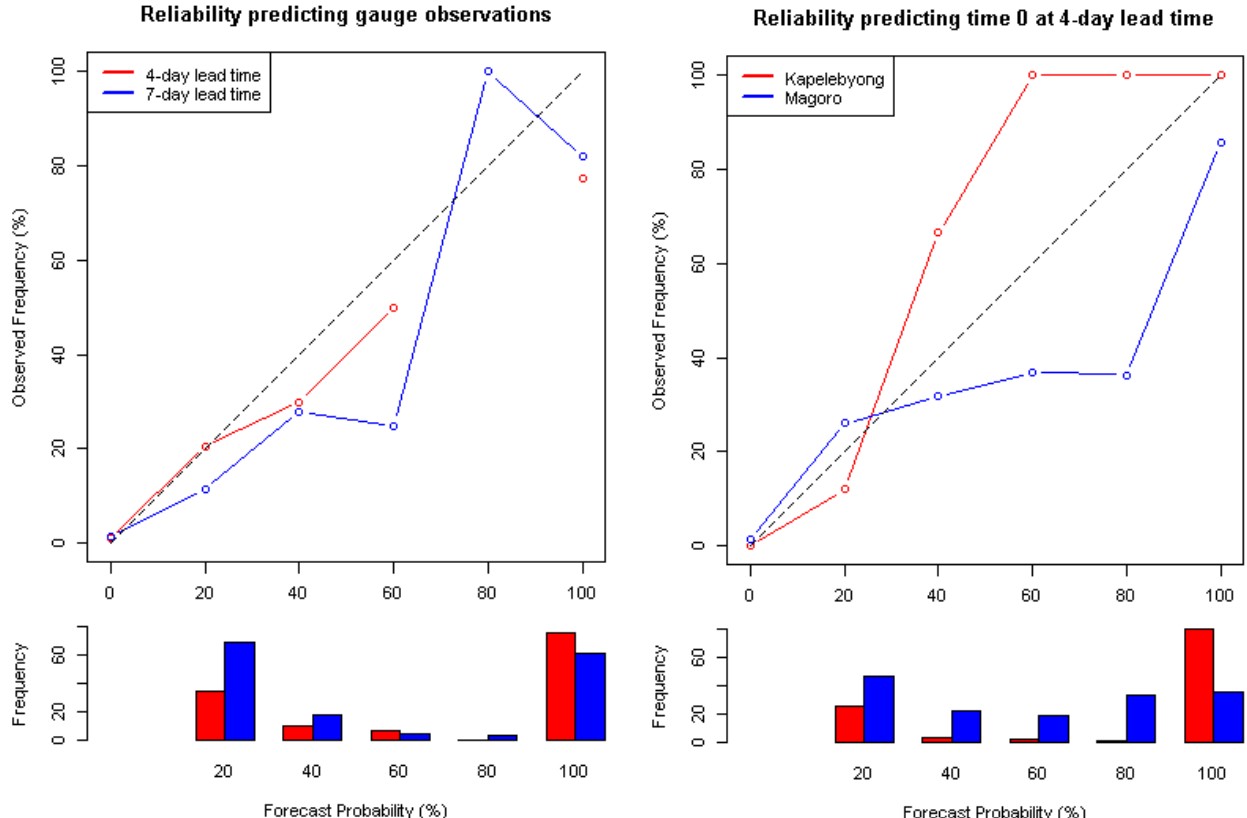

*Figure 3: Reliability diagram for the gauge location (left) and the two project locations (right). This shows how many times a flood occurred for each forecast probability category. For the gauge location, GloFAS forecasts at two lead-times on the Akokoro River are compared with gauge discharge (left). At the two project locations, GloFAS 4-day lead-time forecasts are compared with 0-day forecasts. 7-day forecasts are not shown in the right panel, as results are very similar to the 4-day plots. The frequency of forecast probabilities of 0% are 1688, 1655, 1702, and 1658, for Gauge 4-day, Gauge 7-day, Kapelebyong, and Magoro, respectively. These are not plotted in the frequency bar graph as they would extend past the scale. Lastly, due to sampling uncertainty, 95% confidence intervals extend nearly from 0 to 1, and are therefore not plotted.*





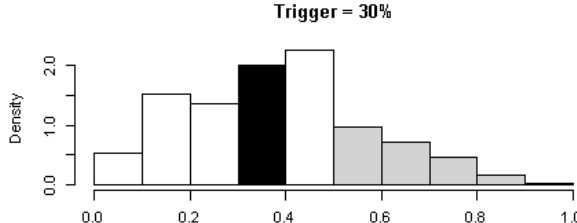

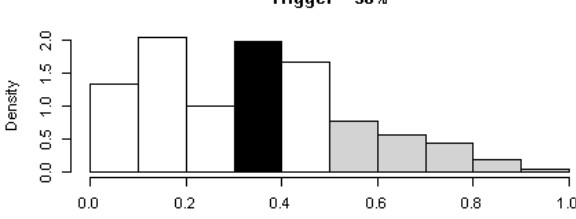

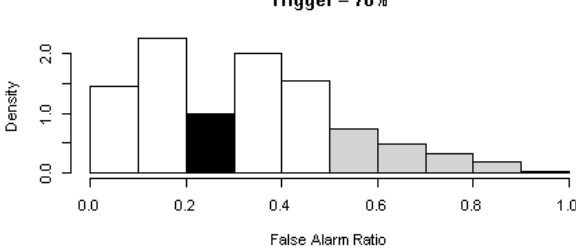

*Figure 4: Histogram of False Alarm Ratio calculations from a block bootstrapped resample of a timeseries of 2014 days of forecast-observation pairs. The vertical axis depicts probability density. Each sample is calculated for 4-day lead times at*

5  *different forecast trigger values. The black bin contains the value of FAR from the original timeseries, and bins exceeding a FAR of 0.5 are grey. All FAR calculations assume a 30-day action lifetime.*