# Peer review of "Action-based flood forecasting for triggering humanitarian action"

_Hydrology and Earth System Sciences, 2016_

## Referee Comment (RC1) · M. Begovic (Referee) · 18 Apr 2016

First of all, big kudos to the team for putting together an excellent research paper with very important development implications. Secondly, i'd like to add two very minor comments that authors may wish to consider when completing the draft:

1. In framing the argument that forecast based finance instrument triggers humanitarian action, i feel that the authors are selling short the overall concept of the mechanism. The characteristics of FbF are such that it corrects what is now a major short coming and a gap between development and humanitarian response, a no-mans land as it were where there is a potential to strengthen resilience of communities that due to the structural make up of dev/humanitarian spaces evolves into a major weakness that sets back development gains ones the event does occur. Therefore i believe the

authors should be more ambitious to claim this space and say that FbF with its pre agreed actions to be committed 'in vain' are strengthening resilience and trigger action that may sit on a very intersection of development (eg. upscaling social services, cash based transfers, irrigation management, etc) and humanitarian actions (eg. digging ditches, bagging food and valuable items, etc).

2. Partly connected to the above, while the paper is about Uganda's case, i believe it may be worthwhile in the closing paragraph to call for a closer collaboration not just between humanitarian and scientific community but int'l development community as well. FbF as a mechanism is a perfect gathering ground for both development and humanitarian sectors to come and plan together.

Thanks for offering an opportunity to review the paper. I've enjoyed it very much. Best of luck,

Millie

---

## Referee Comment (RC2) · Anonymous Referee #2 · 20 Apr 2016

This article illustrates a very interesting example of use of a prototype global forecasting system for taking flood preventive actions, in areas where no alternative flood warning is available. The topic is certainly of high interest, considering that the forecasting system used has global coverage and could be potentially be applied in other regions lacking flood preparedness. The authors are faced with trying to best use the limited amount of ground data available in the region, and devised a clever approach making use of news, media reports, (few) discharge time series, and the output of a global forecasting model. The research issue is well contextualized, methods are rather simple but clear and results are adequately discussed.

As a general comment, I'm surprised not to see some more detail and analysis on the recent event of November 2015 (mentioned on P4, L27 and P14, L21), given that the authors stress the scarcity of data and the limited sample of floods in the observation

period. The current work is based on a relatively small sample of data, making the noise of uncertainty often larger than the actual signal. Hence, an additional event would certainly benefit the analysis.

Further comments

All figures should be cited in the text. This is now not the case for Fig. 1 and 2. Please add a reference.

P7, L14: Flooding is a measure of hazard, not of impact, hence it is independent of exposure and vulnerability. Unless the authors here mean flood risk. Please check.

P8, L1: The clustering algorithm needs a supporting reference.

P8, L5: Not clear how the text in the footnote 2 relates to the flood location. Please clarify.

P8, L11-12: Are these 85-15% obtained by crossing GloFAS forecasts with news report? It's not clear from the text.

The text in page 9 ultimately describes figure 3 and 4 (though with no reference to the two figures). In my opinion this should go to Sect. 4, while Sect. 3 should only include the methodological approach, that is the theory underpinning the FAR, reliability diagram, block bootstrap.

P10, L10: reading through the text it appears that the 95% threshold was chosen after matching the GloFAS data with media reports, rather than an initial qualitative selection.

P13, L21: I suggest rephrasing this, as it currently suggest that the 95% is a general threshold valid for any location. In reality this depends on different factors, not only on the local exposure and vulnerability, but certainly also on the shape of the hydrograph and in turn on the upstream area. This is a consequence of using percentiles in place of extreme value statistics, which are more commonly used for such analyses.

---

## Author Comment (AC1) · 12 Jul 2016

COMMENT: First of all, big kudos to the team for putting together an excellent research paper with very important development implications.

RESPONSE: We thank the reviewer for her positive comments and recommendations.

COMMENT: In framing the argument that forecast based finance instrument triggers humanitarian action, I feel that the authors are selling short the overall concept of the mechanism. The characteristics of FbF are such that it corrects what is now a major short coming and a gap between development and humanitarian response, a no-mans land as it were where there is a potential to strengthen resilience of communities that due to the structural make up of dev/humanitarian spaces evolves into a major weakness that sets back development gains ones the event does occur. Therefore i

believe the authors should be more ambitious to claim this space and say that FbF with its pre agreed actions to be committed 'in vain' are strengthening resilience and trigger action that may sit on a very intersection of development (eg. upscaling social services, cash based transfers, irrigation management, etc) and humanitarian actions (eg. digging ditches, bagging food and valuable items, etc).

Partly connected to the above, while the paper is about Uganda's case, i believe it may be worthwhile in the closing paragraph to call for a closer collaboration not just between humanitarian and scientific community but int'l development community as well. FbF as a mechanism is a perfect gathering ground for both development and humanitarian sectors to come and plan together.

RESPONSE: These are excellent points, and we have made additions to the introduction and conclusion to take this into account. This includes the following sentences:

(Introduction): Indeed, the critical moments in between a forecast and a disaster represent an opportunity to bridge the traditional humanitarian and development spaces. (Introduction): In many cases, pre-agreed actions that are "in vain" because the extreme event did not materialise can have a longer-term positive impact, strengthening resilience and supporting ongoing development efforts in the area. (Conclusion): In particular, the innovation of Forecast-based Financing can encourage the collaboration between development and humanitarian actors to deliberate relevant forecast-based actions; these can both promote and protect long-term development efforts. (Closing paragraph): A closer collaboration between these groups and the international development community should ensure the relevance and success of forecast-based actions.

---

## Author Comment (AC2) · 12 Jul 2016

COMMENT: This article illustrates a very interesting example of use of a prototype global forecasting system for taking flood preventive actions, in areas where no alternative flood warning is available. The topic is certainly of high interest, considering that the forecasting system used has global coverage and could be potentially be applied in other regions lacking flood preparedness. The authors are faced with trying to best use the limited amount of ground data available in the region, and devised a clever approach making use of news, media reports, (few) discharge time series, and the output of a global forecasting model. The research issue is well contextualized, methods are rather simple but clear and results are adequately discussed.

RESPONSE: We thank the reviewer for these comments.

COMMENT: As a general comment, I'm surprised not to see some more detail and analysis on the recent event of November 2015 (mentioned on P4, L27 and P14, L21), given that the authors stress the scarcity of data and the limited sample of floods in the observation period. The current work is based on a relatively small sample of data, making the noise of uncertainty often larger than the actual signal. Hence, an additional event would certainly benefit the analysis.

RESPONSE: This event is currently being studied by the operational team, both in terms of the flooding and the impact of the actions that were triggered. Because all of these results are not yet available or analysed, we have not gone into detail of the event, but we agree it should be mentioned. We have added the following: "while the impacts are still being analyzed, the region reported flooding after the trigger had been reached in one of the project areas."

COMMENT: Further comments All figures should be cited in the text. This is now not the case for Fig. 1 and 2. Please add a reference.

RESPONSE: Thank you for noticing this; it has now been corrected.

COMMENT: P7, L14: Flooding is a measure of hazard, not of impact, hence it is independent of exposure and vulnerability. Unless the authors here mean flood risk. Please check.

RESPONSE: Agreed – we have changed this to read "flood risk".

COMMENT: P8, L1: The clustering algorithm needs a supporting reference.

RESPONSE: Excellent point. We will include the following reference on K-means, as well as a reference to a paper that is currently in preparation: Kaufman, Leonard, and Peter J. Rousseeuw. 1990. Finding Groups in Data; An Introduction to Cluster Analysis. John Wiley & Sons. Hürriyetoglu, A. et al. In prep. [A Tool]: Finding and Labeling Relevant Information in Tweet Collections.

COMMENT: P8, L5: Not clear how the text in the footnote 2 relates to the flood location.

[Figure]

Please clarify.

RESPONSE: We have clarified as follows: "To obtain geographical information, we filtered the sentences for any "marker" terms that are often used when the writer specifies a location, and within this subset we looked for mentions of district and sub-county names."

COMMENT: P8, L11-12: Are these 85-15% obtained by crossing GloFAS forecasts with news report? It's not clear from the text. The text in page 9 ultimately describes figure 3 and 4 (though with no reference to the two figures). In my opinion this should go to Sect. 4, while Sect. 3 should only include the methodological approach, that is the theory underpinning the FAR, reliability diagram, block bootstrap.

RESPONSE: To answer the question about GloFAS, we have added the following clarifying language: "With these results from the algorithm, we validated the result manually for the districts of our interest by reading the articles. For 85% of the events we had found an actual flood event described in the text, meaning that the flood event was automatically detected for the correct month/ year in the correct location(s). Conversely, 15% were false positives, meaning the text was describing a non-flood event."

We prefer to keep the explanation of the newspaper methodology in section 3, as it explains the methods used to obtain the results. We feel it would make the results section more difficult to read if it were placed in section 4.

COMMENT: P10, L10: reading through the text it appears that the 95% threshold was chosen after matching the GloFAS data with media reports, rather than an initial qualitative selection.

RESPONSE: Indeed, the GloFAS data was compared to disaster records both from disaster management agencies and media reports. However, the actual selection of 95 rather than 93 or 97, for example, was qualitative.

COMMENT: P13, L21: I suggest rephrasing this, as it currently suggest that the 95%

is a general threshold valid for any location. In reality this depends on different factors, not only on the local exposure and vulnerability, but certainly also on the shape of the hydrograph and in turn on the upstream area. This is a consequence of using percentiles in place of extreme value statistics, which are more commonly used for such analyses.

RESPONSE: Thank you for noticing this – we have corrected this and it now reads as follows: "Assuming that a specific extreme value of forecasted discharge is a valid proxy for a "danger level" in an area with limited data records, the GloFAS model can be used to trigger timely humanitarian action in advance of an extreme event."